# Navigating the Evolving Landscape of Primary Hyperoxaluria: Traditional Management Defied by the Rise of Novel Molecular Drugs

**DOI:** 10.3390/biom14050511

**Published:** 2024-04-23

**Authors:** Yueqi Huang, Wei Zhu, Jia Zhou, Qiulin Huang, Guohua Zeng

**Affiliations:** 1The First Affiliated Hospital, Hengyang Medical School, University of South China, Hengyang 421001, China; 2023010019@usc.edu.cn (Y.H.); 2018012376@usc.edu.cn (J.Z.); 2Department of Urology and Guangdong Key Laboratory of Urology, The First Affiliated Hospital of Guangzhou Medical University, Guangzhou 510230, China; doczw1989@126.com

**Keywords:** primary hyperoxaluria, oxalate, genetics, RNA interference, *AGXT*, *GRHPR*, *HOGA1*

## Abstract

Primary hyperoxalurias (PHs) are inherited metabolic disorders marked by enzymatic cascade disruption, leading to excessive oxalate production that is subsequently excreted in the urine. Calcium oxalate deposition in the renal tubules and interstitium triggers renal injury, precipitating systemic oxalate build-up and subsequent secondary organ impairment. Recent explorations of novel therapeutic strategies have challenged and necessitated the reassessment of established management frameworks. The execution of diverse clinical trials across various medication classes has provided new insights and knowledge. With the evolution of PH treatments reaching a new milestone, prompt and accurate diagnosis is increasingly critical. Developing early, effective management and treatment plans is essential to improve the long-term quality of life for PH patients.

## 1. Introduction

Oxalate homeostasis controls a major urinary toxin, identified among the top 20, through a complex interplay of synthesis, metabolism, and elimination [1]. Primary hyperoxaluria (PH), a rare autosomal genetic disorder, was discovered and described by Lepoutre in 1925 [2]. This review examines the current biochemical and genetic insights, emphasizing therapeutic progress in PHs. This effort seeks to enhance the comprehension of these disorders.

Oxalate, the simplest dicarboxylic acid (HOOC-COOH), arises both as a metabolic byproduct and from dietary sources, beverages, and chemical constituents [3]. Radiolabeled isotope studies confirm that the kidneys mainly excrete oxalate (>90%), which is a highly insoluble end-product of human metabolism [4,5]. Glomerular filtration and tubular net secretion maintain a physiological oxalate excretion range of 10–40 mg/day (0.1–0.45 mmol/day); however, PH patients exhibit oxalate excretion exceeding 40–45 mg/day (0.45–0.5 mmol/day), accompanied by various systemic manifestations [6,7]. The deposition of calcium oxalate (CaOx) in the renal tubules and interstitium in PH can cause chronic tubulointerstitial inflammation and kidney stone obstruction, resulting in renal failure in over 70% of cases [8].

Primary hyperoxaluria type 1 (PH1), the most prevalent form, has an estimated prevalence of 1–3 individuals per million in Europe, with approximately one case per 120,000 live births [9]. The prevalence data may be conservative, as whole-genome sequencing suggests (Table 1), with an estimated incidence of 1 in 121,499 for PH1, 1 in 196,952 for PH type 2 (PH2), and 1 in 79,499 for PH type 3 (PH3) [10]. In areas with high rates of consanguineous marriage, like North Africa and the Middle East, the PH prevalence can reach 10%, with pediatric stone patients in Pakistan showing a 43% rate of undifferentiated hyperoxaluria [9,11].

## 2. Biochemical Mechanism

The liver primarily synthesizes oxalate from glycolate, as depicted in Figure 1. Cellular oxalate synthesis encompasses three critical phases [8,12]. (1) Mitochondrial collagen turnover and hydroxyproline degradation during animal protein metabolism convert 4-hydroxy-2-oxoglutarate (HOG) into glyoxylate and pyruvate via 4-hydroxy-2-oxoglutarate aldolase (HOGA1). (2) In the peroxisomes, glycolate oxidase (GO) catalyzes the oxidation of glycolate to glyoxylate. Glyoxylate can be transported out or, together with alanine, converted into pyruvate and glycine by alanine-glyoxylate aminotransferase (AGT). Pyruvate can further react with serine in the peroxisomes, forming alanine and hydroxypyruvate, which are then transported to the cytoplasm. (3) Glyoxylate, obtained from mitochondrial catabolism, peroxisomal oxidation, and dietary sources, undergoes conversion to glycolate in the cytoplasm and mitochondria by glyoxylate reductase/hydroxypyruvate reductase (GRHPR), which also transforms hydroxypyruvate into D-glycerate in the cytoplasm. Lastly, lactate dehydrogenase (LDH) in the cytoplasm catalyzes the conversion of hydroxypyruvate and glyoxylate into L-glycerate and oxalate.

PH1 results from a hepatic-specific deficiency in the AGT enzyme, encoded by the *AGXT* gene (Table 1) [13,14]. Lacking AGT, glyoxylate in the peroxisomes fails to be converted into glycine; it is instead transported to the cytoplasm and converted into glycolate and oxalate through the actions of GRHPR and LDH. This newly formed glycolate may be converted back to glyoxylate through the process described in step 2. Consequently, about 75% of PH1 patients exhibit excessive glycolate and oxalate production, while a minority may compensate by elevating the activity of GO, normalizing glycolate excretion [15].

PH2 is caused by a deficiency in the GRHPR enzyme, encoded by the *GRHPR* gene (Table 1) [16]. GRHPR exhibits widespread tissue distribution, primarily within hepatic cells, mostly located in the cytoplasm, with a minor presence in the mitochondria [9]. Due to the deficiency of GRHPR (referenced in step 3), hydroxypyruvate and glyoxylate cannot be converted into D-glycerate and glycolate. Instead, they are metabolized by LDH into L-glycerate and oxalate.

PH3 stems from the functional loss of HOGA1, encoded by the *HOGA1* gene, predominantly found in the liver and kidneys (Table 1) [17]. The loss of HOGA1 function elevates the urine levels of 4-hydroxyglutamate (4OHGlu), 2,4-dihydroxyglutarate (DHG), HOG and oxalate. However, the exact mechanism causing this increase in oxalate levels remains unclear. It is generally postulated that the substrate HOG may be decomposed into oxalate through enzymatic cleavage or the inhibition of the GRHPR enzyme, leading to an increase in oxalate [18,19]. Recent studies propose a third potential mechanism [20]: HOGA1 influences certain metabolic pathways, such as pyruvate availability and the Krebs cycle, increasing the NAD+ concentrations. This increase, in turn, facilitates the conversion of glyoxylate to oxalate by LDH.

## 3. Diagnosis and Genetic Characterization

The diagnosis of PHs typically relies on a combination of clinical symptoms and laboratory data, as illustrated in Figure 2. As previously elucidated, specific biochemical markers facilitate subtype identification: elevated glyoxylate indicates PH1, increased L-glycerate suggests PH2 and heightened 4OHGlu, DHG or HOG indicates PH3. However, the dependability of these biochemical markers for assessment is uncertain. The latest consensus statement on PHs from OxalEurope and the European Rare Kidney Disease Reference Network (ERKNet), published in 2023, emphasizes that genetic testing is the definitive diagnostic gold standard for all PH types [8]. The effective integration of genetic testing and clinical phenotype assessment enhances the application of precision medicine [21]. A daily urine oxalate level exceeding 1 mmol/1.73 m^2^ strongly indicates PH. Before further metabolic or genetic examinations, it is essential to exclude secondary causes of hyperoxaluria (such as chronic pancreatitis, cystic fibrosis, inflammatory bowel syndrome or weight loss surgeries), as these conditions can overlap with the severity of hyperoxaluria seen in PHs [22].

PH1 accounts for approximately 80% of all PH cases (Table 1) [23]. The pathogenic gene *AGXT*, situated on 2q37.3 (MIM# 259900), has been associated with 220 disease-causing mutations (http://www.hgmd.cf.ac.uk, accessed on date 5 April 2024), disrupting AGT function via the loss of catalytic activity, mislocalization, accelerated degradation or protein aggregation [24,25]. The prevalent mutation in PH1 is p.G170R, associated with AGT mislocalization, which accounts for about 28–30% of mutated alleles, primarily found in Western populations [10,26,27]. The relatively common mutations c.33dupC (approximately 12–15%) and p.I244T (approximately 6%) in PH1 exhibit varied frequencies among ethnicities, with p.I244T being more prevalent in Spanish and North African populations and p.G350D showing higher occurrence in those with Pakistani ancestry compared to p.G170R [10,11,28]. PH1, the most severe form of PH, typically presents initial symptoms around the age of 4.9 years, and nearly all such patients advance to end-stage renal disease (ESRD) [29,30]. However, PH1 patients exhibit significant clinical variability, ranging from infantile renal failure with systemic oxalosis symptoms to late-stage renal failure with recurrent urolithiasis and/or nephrocalcinosis, with diverse presentations. PH1 patients with ESRD face a mortality risk about threefold times higher than those without PH1 [31].

PH2 comprises approximately 7.9–10% of all PH cases (Table 1) [7]. The pathogenic gene *GRHPR*, situated on 9p13.2 (MIM# 260000), exhibits common mutations, with c.103delG prevalent in Caucasians (approximately 31–35%) and c.404+3_404+6del and p.G165D mainly observed in Asians (both approximately 14–18%) [10,32]. No disparity in renal functional survival rates has been found among the homozygotes of the three aforementioned most prevalent mutations [32]. PH2 patients typically exhibit a milder clinical phenotype than those with PH1, with the onset of initial symptoms around 5.7 years, but still exhibit a considerable risk of CKD (50%) and ESKD (25%) in adulthood [30,33,34].

PH3 constitutes approximately 8.4–17% of all PH cases (Table 1) [7]. The pathogenic gene *HOGA1*, located on 10q24.2 (MIM# 613616), harbors the most prevalent mutation, c.700+5G>T (accounting for approximately 50%), which causes incorrect mRNA splicing [35,36]. Similar to PH1, PH3 exhibits variations in mutation frequency among different ethnicities. The c.944-46delAGG mutation is more prevalent in those of Ashkenazi Jewish descent, while, in a Chinese cohort study, the c.834_834+1GG>TT mutation accounted for approximately 50% of the mutant alleles [10,37]. PH3, presenting the mildest clinical phenotype among the PH subtypes, typically shows initial symptoms by 2.7 years, mainly characterized by early onset and lifelong nephrolithiasis activity in childhood [30]. However, the severity of PH3 should not be underestimated. Three documented cases in the literature show patients progressing to renal failure at 8, 33 and 78 years, with continuous impairment from oxalate and CaOx crystal deposition, which might be a primary contributing factor [10,38,39]. While most PH3 patients maintain favorable renal function, clinical remission is possible despite persistent hyperoxaluria, emphasizing the need for long-term follow-up support due to the lifelong risk of chronic kidney disease (CKD) progression [20,37].

A distinct category of hyperoxaluria, without a specific subtype, arises from genetic changes affecting intestinal wall function. A patient diagnosed with hereditary enteric hyperoxaluria, manifesting as CaOx renal stones, exhibited a rare missense heterozygous mutation (c.1519C>T/p.r507W) in the gene related to the intestinal secretory oxalate transporter, *SLC26A6*, as identified through genetic testing [40]. The mechanism possibly entails *SLC26A6* inactivation, leading to decreased oxalate transport from the blood to the intestines for excretion, a proposition supported by confirmed findings in mouse models [41]. However, such genetic variants associated with the intestinal wall do not fall under the category of PHs, necessitating differentiation from it.

The clinical phenotype and progression of PHs are contingent upon the severity and type of mutation, and its natural history is typically divided into two stages [7]: (1) initially, elevated oxalate synthesis is compensated for by hyperoxaluria, but, when the urinary oxalate reaches an extreme value of 1–2 mmol/1.73m^2^/day, it causes renal deposition and progressive kidney damage; (2) the subsequent renal function decline impedes oxalate clearance, causing systemic oxalate accumulation and secondary organ damage.

Systemic oxalosis, caused by elevated plasma oxalate (POx), leads to the extensive deposition of CaOx in various tissues, including the bone, retina, myocardium, vascular walls and skin [9]. Hence, genetic evaluation is essential for individuals with suspected PHs, since confirming and identifying the PH gene variants plays a pivotal role in devising appropriate treatment strategies. Preconception counseling and preimplantation genetic testing, coupled with chorionic villus sampling between 10 and 12 weeks, are imperative for couples at an elevated risk of PH to avert severe early-onset disease in their offspring, especially relevant with the advent of RNAi therapies [42,43]. The significance of early PH detection in the siblings of affected individuals has grown with the development of RNAi treatments that mitigate renal damage. Furthermore, with the rise of artificial intelligence (AI) technology, its application across various hematology domains, including digital pathology, cytogenetics, immunophenotyping and sequencing, will further facilitate the early diagnosis of genetic diseases like PHs [44].

## 4. Treatment

In recent years, with rapid advancements in molecular therapy and the continual development of novel pharmaceuticals, the therapeutic strategies for PHs have undergone substantial alterations compared to the past. A succinct depiction of the currently implemented clinical treatments is presented in Figure 2, while further exploration continues to unveil additional therapeutic modalities for the benefit of PH patients.

### 4.1. Related Conservative Treatments

PH patients are recommended to actively increase their fluid intake to urine dilution, critical in preventing the formation of CaOx stones [45]. The European Association of Urology (EAU) advises adults to consume 3.5–4 L of fluid daily, while the latest consensus from OxalEurope and ERK experts increases the fluid intake recommendation for children from 1.5 L/m^2^ of the body surface area (BSA) to 2–3 L, to ensure a minimum urine volume of 2.5 L every 24 h [8,46]. A gastrostomy tube may be required to ensure adequate nocturnal fluid intake for infants unable to self-regulate their fluid intake. Urinary-promoting drugs such as tolvaptan and thiazide diuretics decrease CaOx renal stone formation by reducing the urinary calcium and oxalate levels [47,48]. Furthermore, neutral phosphates (orthophosphates) delay stone formation by reducing CaOx crystallization, recommended at a daily dosage of 20–30 mg/kg phosphate [49,50]. Unfortunately, orthophosphate has never been evaluated independently of other treatments. Similarly, in vitro research has shown that citrate dissolves calcium oxalate monohydrate (COM) crystals by alkalinizing the urine, and a cohort study has demonstrated the benefits of oral potassium citrate at 0.1–0.15 g/kg for PH patients [50,51]. Chung et al. found that hydroxycitrate is a more effective inhibitor of COM nucleation compared to citrate, showing promising clinical potential [52]. Given the gastrointestinal adverse effects of citrate preparations, a 2022 clinical trial suggests that lemon juice could be a viable alternative [53].

Although dietary oxalate absorption is low (5–15%), it contributes 20–50% of the body’s total oxalate content [7].

Although the typical intake of oxalates in a Western diet (100–200 mg/day) is considered harmless for those with normal kidney function, numerous studies have reported cases of acute kidney disease induced by the excessive consumption of oxalate-rich foods like spinach, star fruit, chocolate and nuts [54,55]. The 2023 expert consensus states that PH patients need not adhere to a low-oxalate diet but should limit their intake of foods extremely high in oxalate [8]. Ongoing research is exploring therapeutic interventions targeting the modulation of the gut microbiome to influence oxalate homeostasis. *Oxalobacter formigenes* not only degraded oxalate in the gut but also produced a prosecretory factor enhancing oxalate’s secretion into the intestines in a mouse model of PH1 [56]. Rats pre-treated with *O. formigenes* in a model of ethylene glycol-induced hyperoxaluria and renal calcium deposition showed no significant elevation in POx and the absence of renal calcium deposition compared to the control group [57]. Intestinal capsules of lyophilized *O. formigenes* with a higher live bacterial count, known as OC5 (in contrast to OC3), significantly reduced the POx levels in PH patients [58,59]. Although early research indicated the limited efficacy of Oxadrop (a mixture of four lactic acid bacteria) in a randomized controlled trial to treat hyperoxaluria patients, exploration in this field of research continues unabated [60]. Recent studies reveal that the *Lactobacillus paragasseri* UBLG-36 and *Lacticaseibacillus paracasei* UBLPC-87 bacterial strains, owing to their probiotic properties, prevent hyperoxaluria and attenuate renal injury associated with kidney stones [61,62]. A phase 3, double-masked, placebo-controlled, randomized study evaluating the efficacy of oral Oxabact™, a lyophilized *O. formigenes* formulation, in reducing the POx levels in PH patients found that POx showed stabilization or a reduction in the Oxabact™ treatment group, albeit without statistical significance over 12 months (*p* = 0.06) [63]. The observed subtle effects of Oxabact™ indicate that *O. formigenes* preparations may contribute to preventing kidney stones.

The delivery of enzymes into the gastrointestinal tract by means of oral medication is referred to as enzyme supplementation therapy (EST). Oxalate decarboxylase (ODC) and oxalate oxidase (OxO), the oxalate-degrading enzymes under investigation for drug development, degrade oxalate into human-safe products, despite both originating from non-mammalian proteins [64]. Although the development of such therapeutic drugs (such as Nephure™, OxDc-CLEC, Oxazyme and ALLN177) was aimed at patients with oxalosis and secondary hyperoxaluria, their suggestive role in the treatment of PH cannot be denied [64]. OxDc-CLEC has demonstrated effectiveness in reducing hyperoxaluria and preventing CaOx nephrocalcinosis and urolithiasis in hyperoxaluria mouse models, independently of the etiology [65]. Unfortunately, in a clinical trial of Oxazyme (OC4; ClinicalTrials.gov number, NCT01127087), only patients with enteric hyperoxaluria post-Roux-en-Y gastric bypass showed improved efficacy, not those with idiopathic hyperoxaluria. In a phase 1 clinical trial (NCT02289755), ALLN177 significantly reduced the 24-h urinary oxalate (UOx) excretion among five subjects with enteric hyperoxaluria and 11 with idiopathic hyperoxaluria [66]. Following these encouraging findings, a subsequent clinical study of ALLN-177 involving patients aged 12 years or older with enteric or primary hyperoxaluria and hyperoxaluria (NCT03391804) was conducted, completed in late 2019, although the results remain unpublished.

Certain metallic cations also possess the capacity to modulate oxalate metabolism. Magnesium (Mg) can inhibit not only CaOx crystallization in human urine but also the absorption of dietary oxalates from the gut lumen [67]. Clinical trial evidence does not support the use of magnesium as a monotherapy for CaOx kidney stones; however, Gheissari et al.’s study demonstrated that magnesium combined with citrate surpassed citrate alone in efficacy for pediatric urolithiasis patients [67,68]. Similarly, magnesium salts effectively prevented stone formation in a rat model of calcium oxalate kidney stones, with Mg aspartate and Magne B6 demonstrating superior efficacy compared to other studied salts [69]. Lanthanum carbonate (LaC), a rare earth metal with low bioavailability (0.00127 ± 0.0008%) and predominant biliary excretion (99%), attenuates UOx levels by sequestering oxalates within the intestinal milieu, as validated in a rat model of hyperoxaluria [70,71]. Based on this evidence, Agnieszka et al. applied Lac therapy in two PH1 patients with distinct conditions (patient 1 in the anuric ESRD stage with systemic oxalosis and patient 2 with normal renal function), achieving a decrease in POx to below the target value, indicating Lac’s potential therapeutic value for PH patients [72]. Wu et al.’s 2023 study demonstrated that zinc gluconate could markedly augment the abundance of oxalate-metabolizing bacteria in humans, thereby alleviating symptoms in a CaOx kidney stone rat model [73]. The relevant studies in this domain have provided novel insights and directions for future therapeutic research on PHs.

### 4.2. Chaperone Therapy

Vitamin B6 (pyridoxine), metabolized in the body to pyridoxal 5′-phosphate (PLP; biologically active form), functions as a critical cofactor and chaperone for AGT enzymes, restoring lost enzyme function in PH1 through mechanisms related to increased expression, catalytic activity and peroxisomal input to AGT enzymes [74,75]. Given the risk of neurotoxicity from prolonged and high-dose pyridoxine administration, the recommended maximum intake is 5 mg/kg, and patients requiring higher doses should be closely monitored [8]. Pyridoxine reactivity is defined as an average UOx decrease exceeding 30% across at least two consecutive measurements following a minimum of 2 weeks of continuous administration in PH1 patients [76]. The response to pyridoxine among different PH1 patients exhibits substantial individual diversity, with an estimated 10–20% achieving the complete normalization of UOx excretion, 30% showing a partial response and 50–60% displaying no response [77]. Pyridoxine reactivity testing is essential, enabling some PH1 patients to possibly forego or postpone expensive treatments like RNAi therapy or invasive procedures, notably organ transplantation, a key consideration in low-income countries [78]. Currently, there is substantial and reliable clinical evidence for the in vivo reactivity of the p.G170R-Mi and p.F152I-Mi mutations [79,80]. Additionally, relevant in vitro experiments have validated the p.G47R-Mi, p.156N-Ma and p.G161R/S/C mutations, indicating their potential for in vivo reactivity [24,81,82]. Other vitamin B6 forms, pyridoxamine (PM) and pyridoxal (PL), demonstrate superior efficacy to PLP in certain mutants (p.G170R-Mi, p.F152I-Mi, p.G41R-Ma, p.G161R-Ma and p.I244T-Mi) [78,83]. However, this effectiveness has so far been confirmed only through in vitro studies, with future clinical research anticipated.

Betaine (trimethylglycine), a modified amino acid, mainly functions to maintain osmotic pressure, protect denatured proteins as a “chemical chaperone” and detoxify homocysteine [84]. Betaine demonstrated a stabilizing protective effect against specific pathogenic AGT variants (p.G170R-Mi, p.I244T-Mi and p.F152I-Mi) in vitro PH cell models [85,86,87]. Despite betaine’s well-tolerated nature in PH1 patients (n = 10), as the sole companion drug to VB6 subjected to clinical trials (NCT00283387), it showed no significant reduction in UOx levels. The results of the aforementioned clinical trial remain unpublished, necessitating future research to assess betaine’s efficacy through formulation or dosage adjustments and expanded sample sizes. Beyond VB6, dequalinium chloride (DECA) and its translation elongation inhibitor, emetine, originally employed clinically for antimicrobial therapy in the oral and vaginal domains, as well as for the treatment of amoebiasis, have also been demonstrated in in vitro experiments to be drugs capable of rectifying the mitochondrial mislocalization of AGT [88,89]. However, the specific therapeutic efficacy of both requires further confirmation through clinical validation.

### 4.3. Dialysis Treatment

PH patients at CKD stage 4–5, even without progressing to ESRD, require assessment for systemic oxalosis risk to evaluate the need for dialysis therapy. Therefore, patients with persistently elevated urinary oxalate levels despite oxalate-lowering treatments (such as pyridoxine or RNAi) should begin dialysis early due to the primary threat of oxalate accumulation in advanced CKD stages [8]. High-flux filters for enhanced hemodialysis (HD) are essential, as conventional dialysis regimens cannot adequately counteract the elevated oxalate production in PH patients [90]. The formulation of an enhanced dialysis regimen should involve an increase in the frequency of dialysis sessions per week, rather than extending the duration of each session, as the efficacy of oxalate clearance during HD diminishes with the reduction in POx levels throughout the dialysis process [90]. When tolerated, patients should undergo daily HD therapy, with the option of incorporating nocturnal peritoneal dialysis if necessary, to maintain pre-dialysis POx levels below 30–45 μmol/L in PH patients [9]. However, even combined high-intensity dialysis modalities cannot eliminate the accumulation of oxalates or reverse oxalate tissue deposition, regularly serving as a supplementary measure or a bridge leading to the ultimate therapeutic approach [12].

### 4.4. Surgical Interventions

Surgical interventions for PH-related kidney stones are conventional, but, given the rarity of the disease, special consideration is warranted for treatment indications and potential side effects [91]. PH patients rarely undergo open or laparoscopic access, particularly transparenchymal approaches, due to the higher invasiveness. Given that the predominant stone component in PHs is COM, shock wave lithotripsy (SWL) is challenging due to its inefficacy in fracturing COM stones, which are difficult to fragment using this technique, resulting in a low treatment success rate for PH1 patients and up to 61% requiring repeated interventions [92,93]. PH2 patients exhibit a better therapeutic response to SWL than PH1 patients, possibly attributed to this subset’s composition of mixed oxalate–phosphate stones [92]. For all PH patients, initial ureteroscopy (URS) is advised to evaluate the actual stone burden, making URS with lithotripsy the preferred first-line treatment for those with stone burdens <20–30 mm [94,95]. Amidst a substantial stone burden, the surgical approach can smoothly transition to percutaneous nephrolithotomy (PCNL) within the same anesthetic session, reducing the number of patients undergoing unnecessary, highly invasive procedures [91]. Managing kidney stones in PH patients requires a delicate balance: surgery does not guarantee a prolonged stone-free status, while stone obstruction can acutely impair the estimated glomerular filtration rate (eGFR), disrupting the oxalate homeostasis and risking further renal function deterioration [9,96]. Hence, urologic clinicians must carefully account for PH patients’ unique aspects compared to other stone-forming groups in devising treatment plans.

Liver transplantation (LT) stands as the only definitive cure for PH1, reversing hyperoxaluria and stopping the progression of oxalate-related disorders [12]. For PH1 patients with renal failure, combined liver–kidney transplantation (CLKT) is recommended, with a series of studies demonstrating superior renal transplant survival rates compared to isolated kidney transplantation (KT) [31,97,98]. There is an ongoing debate about the comparative merits of CLKT and sequential liver–kidney transplantation (SLKT) for PH patients, with three high-quality studies revealing no significant differences in the overall survival and renal transplant survival rates between the two approaches [99,100,101]. A 2023 study on long-term outcomes for infants and adolescents with PH1 (n = 18) undergoing CLKT/SLKT showed encouraging results, with a 94% patient survival rate (median follow-up of 9.2 years) and liver and kidney transplant survival rates reaching 85% and 75%, respectively, after 15 years [102]. However, pre-emptive liver transplantation (PLT) as a transplant strategy remains controversial. Data from the OxalEurope registry indicated that PH1 patients (n = 12) undergoing PLT had a higher mortality rate (16.7%) and a higher risk of progression to ESRD (16.7%) post-treatment [103]. Nevertheless, a retrospective study published in 2023 demonstrated that PH1 patients undergoing PLT had better overall survival and excellent kidney outcomes during long-term follow-up [104]. Historically, isolated KT was considered suitable only for specific PH1 patients, notably those with pyridoxine sensitivity and normal or near-normal UOx excretion [103]. However, the recent application of RNAi therapy in PH patients has gradually broadened the prospects for the clinical use of isolated KT. The first case report describes a 39-year-old PH1 female who achieved POx normalization before isolated KT, following three years of conventional HD and one year of daily HD with seven months of Lumasiran treatment [105]. Subsequently, Anne-Laure et al.’s case reports demonstrated that combining isolated KT with Lumasiran in five PH1 patients resulted in successful outcomes for all during a follow-up period of at least six months [106]. In another case, a 5-year-old female PH1 patient progressing to renal failure had her POx decreased to 55 mmol/L after Lumasiran and Nedosiran treatment, ultimately undergoing successful isolated KT with a favorable outcome at the 5-month follow-up [107]. In conclusion, enhancing KT’s success in PH patients relies on the sustained long-term control of POx and the avoidance of CaOx crystal deposition [108]. Future hopes rest on obtaining abundant and comprehensive long-term clinical data to guide clinicians in developing diagnostic and therapeutic strategies.

PH2 patients exhibit a milder clinical phenotype compared to PH1 patients, and, correspondingly, a clinical transplant data scarcity exists. A case report published in 2014 revealed isolated KT therapy’s failure in a pediatric PH2 patient [109]. Similarly, an adverse outcome in a 33-year-old PH2 patient following isolated KT suggests CLKT as a potentially better option [110]. A 2019 study based on OxalEurope data for PH2 patients treated with isolated KT showed that out of 10 patients (after excluding two lost to follow-up), four out of eight received two kidney transplants, of which two patients died in the third and fifth years after the repeat KT; one patient who underwent CLKT faced a primary graft insufficiency and died from sepsis one year later [32]. In a case reported in 2022, a 26-year-old PH2 patient progressing to ESRD experienced oxalate nephropathy recurrence, confirmed by renal biopsy on the 33rd day post-KT; regrettably, subsequent LT failed to reverse the situation [111]. Arnaud et al. reported an illustrative case: a 41-year-old male PH2 patient experienced the rapid and severe recurrence of oxalate nephropathy after isolated KT, leading to early graft loss, yet he was subsequently re-treated with CLKT without relapse and with successful resolution [112]. In the past few years, two successful CLKT treatments in PH2 patients have been reported: a 44-year-old man with frequent stone events and ESRD, and a 12-year-old boy with systemic oxalosis and ESRD [113,114]. The deficient enzyme in PH2 lacks liver specificity, historically suggesting isolated KT for such patients [9]. However, clinical data reviews indicate this approach’s poor prognosis, marked by low graft survival rates and high complication incidences. Therefore, for PH2 patients, transplant strategies incorporating combined CLKT or KT following LT may be more suitable. Nonetheless, establishing reliable guidelines requires the collection of comprehensive data from a large cohort of post-transplant PH2 patients.

### 4.5. Substrate Reduction Therapy (SRT)

In recent years, newly developed RNAi therapies for PHs have focused on the therapeutic targets of GO and LDH. Identifying relevant targets relies on their roles in key oxalate metabolism steps and seeks to minimize off-target effects. Researchers continuously enhance RNAi drugs’ chemical modifications to boost their metabolic stability, specificity and efficacy and reduce treatment immunogenicity [115].

Lumasiran (Oxlumo^TM^) is a synthetic double-stranded RNAi targeting the mRNA encoding the GO enzyme of hydroxy acid oxidase 1 (*HAO1*) [116]. This drug has received marketing authorization from the European Medicines Agency (EMA) and the U.S. Food and Drug Administration (FDA) as an orphan drug for PH1. Administered subcutaneously, Lumasiran effectively targets the liver and minimizes systemic off-target effects via a high affinity for hepatic asialoglycoprotein receptor 1 [117,118]. An investigation showed that mice with a GO deficiency had elevated urinary glycolate without other phenotypes, affirming the safety and efficacy of targeting GO as a therapeutic intervention [117,119]. Frishberg et al. reported two 8-year-old brothers with HAO1 homozygous mutations from parental consanguinity, showing extremely high urinary glycolate levels since infancy, without hyperoxaluria or kidney stones, thus providing clinical support for the subsequent development of SRT therapies targeting HAO1. The results from a placebo-controlled randomized phase 1/2 clinical trial (NCT02706886) showed Lumasiran’s acceptable safety and normalized urinary excretion in PH1 patients [120]. Group B continued to participate in a long-term, open-label extension study (NCT03350451) due to an insufficient patient number and a limited administration duration. Completed on 7 February 2023, the trial’s results are pending submission. Building on the positive outcomes from previous studies, a 6-month randomized, double-masked, placebo-controlled phase 3 trial (Illuminate-A; NCT03581184) reported Lumasiran’s efficacy and safety versus a placebo in PH1 patients aged six and above [121]. The results indicated that Lumasiran normalized or nearly normalized most patients’ levels within six months, with 38% experiencing mild transient injection-site reactions. A single-arm, open-label phase 3 study (Illuminate-B; NCT03905694) evaluated Lumasiran’s efficacy and safety in PH1 patients under the age of 6 with eGFR >45 mL/min/1.73 m^2^ [122]. The results demonstrated Lumasiran’s rapid and consistent reduction of the urinary oxalate to creatinine ratio (UOx:Cr) in PH1 patients <6 years, with an average 72% reduction, and 50% achieved UOx:Cr ≤1.5 times the upper limit of normal by month 6, demonstrating Lumasiran as a safe and effective treatment option for infants and young children. EP, with a study duration >12 months, demonstrated sustained reductions in UOx:Cr across all weight ranges, along with trends towards improved nephrocalcinosis grades and eGFR, but longer-term follow-up data (EP up to 54 months) are still required for additional support [123]. Furthermore, with its widespread clinical application, two PH1 patients reported in 2023 achieved positive outcomes, ceasing nocturnal overhydration at ages 12 and 15 after receiving Lumasiran treatment, indicating that the discontinuation of nighttime fluid supplementation may be safe for children responding to Lumasiran [124]. However, updates to the nighttime hydration treatment guidelines await further RNAi therapeutic data. Meanwhile, an open-label phase 3 trial (Illuminate-C; NCT04152200) is assessing Lumasiran’s efficacy and safety in advanced PH1 patients, including those undergoing HD [125]. The results showed a 42% average POx decrease (95% CI, 34.2%–50.7%) after six months of Lumasiran in advanced PH1 dialysis patients, with an acceptable safety profile. Lumasiran’s efficacy and safety have been validated across various severities of PH1 in the aforementioned clinical trials, reducing the dialysis needs by correcting the hepatic oxalate imbalance. It can improve or prevent the development of systemic oxalosis and potentially reduce the demand for LT in PH1 patients.

Another RNAi therapeutic target is *LDHA*, the mRNA-encoding LDH enzyme. As this target is key in oxalate production’s final step, it holds promise as a critical therapeutic approach for the treatment of all types of PH. Nedosiran, a synthetic double-stranded RNAi targeting LDHA, successfully reduced oxalate production in a hyperoxaluric mouse model during preclinical trials in 2018 [126]. The first-in-human phase 1 study of PHYOX1 (NCT03392896), the initial phase 1 study of Nedosiran, assessed its safety, tolerability, pharmacokinetics and pharmacodynamics in two groups: Group A (25 healthy participants) and Group B (PH1 or PH2 patients) [127]. Group B patients exhibited an average maximum reduction of 55% (range: 22%–100%) in Uox at the end of treatment, with 33% achieving normal 24-h UOx excretion. Both groups did not exhibit any severe safety issues. Based on simulated data, a safe and effective dose of Nedosiran for adults aged ≥18 years was determined to be 160 mg/month, paving the way for subsequent studies [127]. PHYOX2 (NCT03847909), a randomized, double-masked, placebo-controlled study, investigated Nedosiran’s efficacy and safety in participants with eGFR ≥30 mL/min/1.73 m^2^, including PH1 (n = 29) and PH2 (n = 6) [128]. The results revealed a sustained UOx reduction and improvements in POx and the stone burden in the PH1 subgroup during treatment, whereas the PH2 subgroup did not exhibit consistent results. PHYOX4 (NCT04555486), a randomized, double-blind, placebo-controlled phase 1 study, assessed a single Nedosiran dose’s safety and tolerability in PH3 patients [129]. The results indicated a mean UOx reduction of 24.5% in patients treated with Nedosiran (falling short of the goal of a 30% reduction), with an acceptable and well-tolerated safety profile. Nedosiran clinical studies are currently limited by short trial durations and a lack of application in severely renal-impaired patients and younger age groups. PHYOX3 (NCT04042402), PHYOX7 (NCT04580420) and PHYOX8 (NCT05001269) are actively addressing these issues, with anticipated completion dates in April 2030, May 2025 and November 2023, respectively. Of note, the efficacy of Nedosiran in PH2 and PH3 patients still needs to be further explored.

Stiripentol is typically employed for the treatment of genetic epileptic encephalopathy Dravet syndrome [130]. In 2015, Sada et al. demonstrated that Stiripentol targeted the LDH5 isoenzyme in neuronal in vitro studies [131]. Marine et al. (2019) confirmed that Stiripentol inhibited oxalate formation in human liver cells and live mice by targeting the LDH5 isoenzyme, offering new hope for the prevention of the kidney and systemic adverse effects caused by PHs [132]. The team achieved notable success by administering Stiripentol to a 17-year-old girl with PH1 and her brother, who was also diagnosed with PH1 and underwent a CLKT, resulting in a significant reduction in UOx levels [132]. Stiripentol treatment also normalized the UOx:Cr levels and dissolved kidney stones in an 18-month-old child with PH1 [133]. Nevertheless, the compassionate therapeutic effects of Stiripentol in two PH1 patients, CKD (Patient 1) and ESRD (Patient 2), did not align with the results of the LDHA-RNAi study (NCT03392896), indicating an inadequate reduction in oxalate production in these individuals [134]. Following a compassionate treatment for 4 months with Stiripentol in a dialysis-dependent infant with PH1, the POx levels did not significantly decrease, suggesting limited efficacy in advanced CKD PH patients [135]. The first case report on Stiripentol in an adult PH1 patient indicates that combining it with Lumasiran for early oxalate nephropathy relapse post-KT is safe and effective, potentially offering an alternative to CLKT for PH1-associated advanced CKD [136]. Thus, Stiripentol’s efficacy appears contingent upon the renal function of PH patients, suggesting that only patients with preserved renal function in the early stages will benefit from the drug. Nonetheless, larger-scale studies are required to validate this conclusion. The open-label phase 2 study of Stiripentol (NCT03819647), which was completed in March 2021, was conducted to investigate the drug’s efficacy as a monotherapy in PH patients older than six months (eGFR > 45 mL/min/1.73 m^2^), but the relevant results have not yet been published. The LDH small-molecule inhibitor CHK-336 demonstrated dose-dependent UOx level reduction and normalization in a PH1 mouse model, and, as of April 2023, the phase 1 clinical study (NCT05367661), which included a placebo control, single dose and multiple escalating doses, has been concluded, with the relevant results pending release [12].

Research has also investigated LDH’s selective inhibition by 20 compounds with a 2,8-dioxabicyclo[3.3.1]nonane scaffold [137]. Notably, some compounds showed oxalate production inhibition in PH1-3 cell models, matching or exceeding Stiripentol’s effect. This offers a basis for structural optimization in developing new molecular therapeutics for PHs. Nicolás et al. developed a GO inhibitor model utilizing QSAR, molecular docking and molecular dynamics simulation methods and subsequently identified seven commercially available drugs within the model’s scope through the DrugBank database, which were tested as potential clinical GO inhibitors for future applications or as lead compounds in the design of therapeutics for PH1 [138]. Artificially synthesized dual inhibitors targeting both the GO and LDH enzymes, including salicylic acid derivatives, bring hope to PH patients, yet the potential synergistic effects of dual-target inhibition are uncertain, and challenges such as low bioavailability and limited tissue specificity present significant hurdles on the path to clinical application [139,140].

### 4.6. Preclinical Therapeutic Explorations

#### 4.6.1. CRISPR/Cas9

Employing viral vectors (adenovirus or lentivirus) or non-viral delivery systems (such as lipofectamine and cell-penetrating peptides) for clustered regularly interspaced palindromic repeats (CRISPR)/Cas9 delivery constitutes a powerful gene editing mechanism for in vivo application [141,142,143]. Contrasted with transient gene silencing technologies like RNAi, this technique, relying on the Cas9 endonuclease’s precision in targeting specific genomic loci, can induce insertions and deletions in the genome, presenting a permanent advantage. This technology has been successfully applied in PH1 animal models. Rui et al. utilized CRISPR/Cas9 technology to specifically target and knock out *HAO1* in 30% of liver cells in a PH1 rat model therapeutic group, achieving a 42% reduction in UOx levels compared to the control group and preventing widespread renal calcification for at least one year [144]. In the same year, a published study utilizing CRISPR/Cas9 to specifically knock down *LDHA* in 20% of hepatocytes in a PH1 rat model therapeutic group resulted in a 50% reduction in LDH expression relative to the control group [145]. This led to a sustained decrease in endogenous oxalate production and reduced renal CaOx precipitation, while maintaining an acceptable safety profile without off-target effects in genomes or organs. The 2023 investigation highlighted the multiple genome editing capabilities of CRISPR from *Prevotella* and *Francisella* 1 (Cpf1), demonstrating that the simultaneous targeting of HAO1 and LDHA in a PH1 rat model effectively reduced the UOx levels, preserved renal function and minimally decreased renal CaOx deposition [146]. Disappointingly, no cumulative effect was observed in this multiple-targeting study, possibly because *HAO1* and *LDHA* are present in the same oxalate metabolism pathway. Enhancing the therapeutic efficacy may necessitate expanding the population of effectively edited hepatocytes. Due to the significant drawbacks, such as off-target effects, challenges in delivery efficiency, high-dose toxicity and the activation of host immune responses, CRISPR/Cas9 technology is currently confined to preclinical trials in vitro cell cultures and animal models [147,148,149]. For CRISPR/Cas9 to be viable in clinical PH treatment, it is crucial to rigorously mitigate both the immediate and unforeseeable long-term risks.

#### 4.6.2. ERT

The current preclinical trials on direct enzyme replacement therapy (ERT) focus primarily on the restoration of AGT enzymes in the hepatic cells of PH1 hosts. Conjugating AGT with poly(ethylene glycol)-co-poly(L-glutamic acid) (PEG–PGA) block co-polymers to create PEG–PGA–AGT conjugates facilitates their smooth internalization and accurate subcellular localization for glyoxylate metabolism in PH1 cell models, while also addressing plasma stability and immunogenicity concerns [150]. AGT gene delivery employs recombinant adeno-associated virus (AAV) vectors with complementary DNA (cDNA) and lipid nanoparticles encapsulating mRNA, effectively reducing the UOx levels in *AGTX* gene knockout mouse models [151,152,153]. Indirect ERT primarily involves liver cell transplantation (LCT) with normal function, either autologous or allogeneic. Autologous LCT requires the prior in vitro restoration of AGT. Julie et al. successfully restored the AGT expression in induced pluripotent stem cells (iPSCs) derived from PH1 patients, which failed to retain the edited AGT during differentiation into hepatocyte-like cells [154]. Subsequent research has achieved the stable expression of AGT in liver-specific differentiated PH1-iPSCs by employing a liver-specific thyroid hormone-binding protein promoter [155]. Allogeneic transplantation was reported in a 15-month-old female with systemic oxalate deposition. Despite a significant POx decrease after LCT from a male newborn donor, biopsy puncture samples taken at the 5th month (Segment 6) and 12th month (all segments) post-LCT showed no detection of transplanted male donor cells [156]. The proliferation of corrected hepatocytes, constituting only a tiny fraction of the total host liver, poses a primary challenge for indirect ERT. While liver X-ray irradiation and liver cell growth factor stimulation have proven effective in a PH1 mouse model, these methods are not suitable for human PH patients [157].

#### 4.6.3. Immunoregulation

Oxalate induces various inflammatory pathways in the kidney, contributing to PH progression. NLRP3 (NOD-, LRR- and pyrin domain-containing protein 3), acting as a cellular sensor, forms and activates its inflammasomes, leading to the downstream maturation of pro-inflammatory factors IL-1β and IL-18, thereby triggering inflammation and mediating the cellular death (pyroptosis) executed by gasdermin-D [158,159]. Anakinra, an IL-1β receptor antagonist approved by the FDA to treat rheumatoid arthritis, also exerted a protective effect against inflammation and tissue damage during CaOx crystal-induced kidney injury in a mouse model [160]. In a renal tubular injury cell model established by processing human renal proximal tubular epithelial cells (HK-2) with COM, miR-22-3p impeded the release of pro-inflammatory factors and the generation of cellular pyroptosis through the inhibition of NLRP3 transcript levels [161]. CP-456,773, an inhibitor of inflammatory vesicles, reduced renal inflammation and fibrosis by targeting NLRP3 in a mouse crystal nephropathy model induced by an oxalate-/adenine-rich diet [162]. Tumor necrosis factor-alpha receptor (TNFR) signaling mediates inflammatory responses and necrotic apoptosis in renal crystal deposition and crucially influences CaOx crystal adhesion to renal tubular luminal membranes [163,164]. R-7050, a small molecule inhibiting TNFR1 and TNFR2 signaling, prevented renal calcinosis and CKD progression in a hyperoxaluria-induced mouse model (induced by an oxalate-rich diet) [164].

## 5. Conclusions

For an extended period, PHs have been characterized by their rarity, diverse clinical presentations and complex diagnosis, posing a significant challenge for healthcare professionals, particularly in developing countries. This review provides a comprehensive summary and synthesis of the distinctive characteristics of various subtypes of PH, aiming to equip clinicians with a thorough understanding for precise and timely diagnosis, the cornerstone of precision medicine. The continuously evolving therapeutic landscape for PHs holds significant potential in markedly reducing hepatic oxalate production. It is anticipated that the future demand for organ transplantation will substantially decrease, given the diminished proportion of patients experiencing advanced CKD or ESRD. As the diagnosis and treatment of PHs continue to advance and be refined, the quality of life of subsequent generations of PH patients will hopefully be transformed.

## Figures and Tables

**Figure 1 biomolecules-14-00511-f001:**
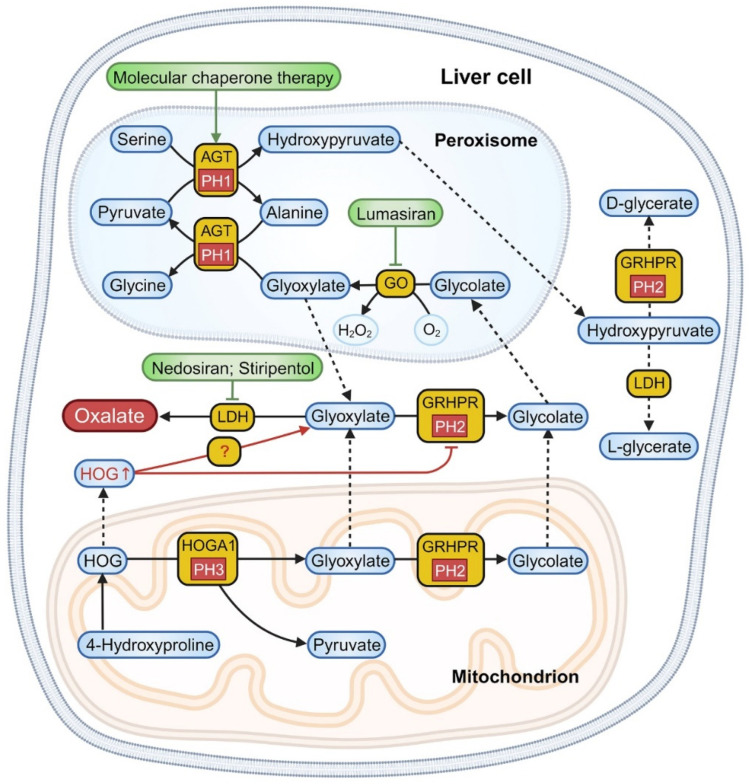
Oxalate metabolism pathways and related therapeutic targets in hepatic cells. PH1 arises from a hepatic-specific deficiency in the AGT enzyme. PH2 arises from the scarcity of the GRHPR enzyme, mainly situated in hepatic cells, predominantly located in the cytoplasm of hepatic cells, with a lesser presence in mitochondria. PH3 arises from the functional loss of HOGA1, typically leading to hyperoxaluria through the inhibition of the GRHPR enzyme and the decomposition of the substrate HOG via an unknown pathway. Contemporary pharmacotherapy for PHs typically focuses on targeting the AGT, GO and LDH enzymes and their associated encoding genes. PHs: primary hyperoxalurias; PH1–3: primary hyperoxaluria type 1–3; AGT: alanine-glyoxylate aminotransferase; GO: glycolate oxidase; GRHPR: glyoxylate reductase/hydroxypyruvate reductase; LDH: lactate dehydrogenase; HOG: 4-hydroxy-2-oxoglutarate; HOGA1: 4-hydroxy-2-oxoglutarate aldolase. The figure was created with BioRender.com.

**Figure 2 biomolecules-14-00511-f002:**
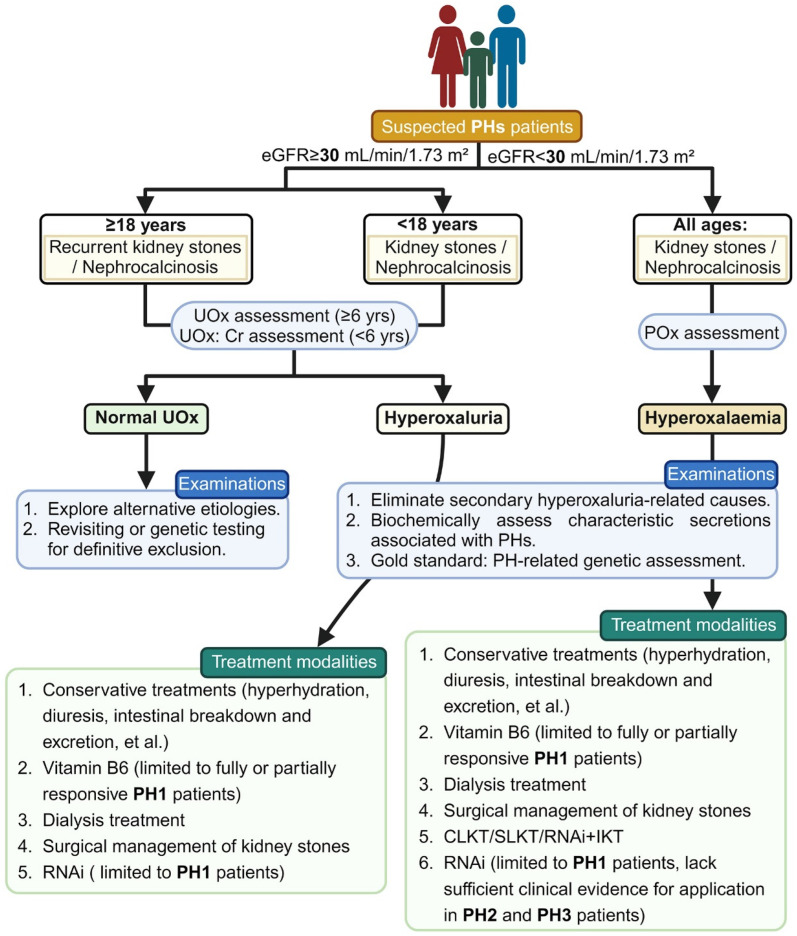
The diagnostic and therapeutic protocol for PH patients. PHs: primary hyperoxalurias; PH1–3: primary hyperoxaluria type 1–3; eGFR: estimated glomerular filtration rate; UOx: urinary oxalate; Cr: creatinine; POx: plasma oxalate. RNAi: RNA interference. The figure was created with BioRender.com.

**Table 1 biomolecules-14-00511-t001:** Synopsis of clinical features associated with PHs.

Clinical Features	PH1 (Approximately 80%)	PH2 (Approximately 7.9–10%)	PH3 (Approximately 8.4–17%)
Estimated incidence rate	1:121,499	1:196,952	1:79,499
Affected enzyme	AGT	GRHPR	HOGA1
Enzyme distribution	Hepatic-specific distribution (peroxisome)	Predominantly distributed in liver (predominantly in the cytoplasm and a smaller portion in mitochondria)	Predominantly distributed in liver and kidneys (mitochondria)
Coding gene (chromosomal position)	*AGXT* (2q37.3)	*GRHPR* (9p13.2)	*HOGA1* (10q24.2)
Biochemical markers	Glyoxylate	L-glycerate	4OHGlu, DHG, HOG
Age of initial symptom onset	4.9 years	5.7 years	2.7 years
Clinical performance	Infantile oxalosis, nephrocalcinosis, systemic oxalosis and ESRD	Nephrocalcinosis, systemic oxalosis and ESRD (approximately 25%)	Recurrent calcium oxalate renal stones and ESRD (case reports)

PHs: primary hyperoxalurias; PH1–3: primary hyperoxaluria type 1–3; AGT: alanine-glyoxylate aminotransferase; GRHPR: glyoxylate reductase/hydroxypyruvate reductase; HOGA1: 4-hydroxy-2-oxoglutarate aldolase; 4OHGlu: 4-hydroxyglutamate; DHG: 2,4-dihydroxyglutarate; HOG: 4-hydroxy-2-oxoglutarate; ESRD: end-stage renal disease.

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
