# Peer review of "Navigating the Evolving Landscape of Primary Hyperoxaluria: Traditional Management Defied by the Rise of Novel Molecular Drugs"

_biomolecules, 2024, doi:10.3390/biom14050511_

Round 1
Reviewer 1 Report
Comments and Suggestions for Authors
The manuscript by Huang et al. covers the pathogenesis, clinical features and therapeutic options of primary hyperoxalurias, rare genetic disorders of glyoxylate metabolism. Although the content is timely and comprehensive, the manuscript suffers from shortcomings that need to be addressed before consideration for publication.
1- the use of siRNA as new therapeutic option is described at the end of the manuscript although it is cited before in several instances (i.e. lines 215 and 314 where the new drugs are cited without a prior explanation). I would suggest to remove the references and describe the changes in the therapeutic approach in paragraph 4.5.
2- the format of table 1 should be improved
3- line 132: please check the actual number of mutations
4- line 178: I would not consider SLC26A6 mutation as a form of primary hyperoxaluria
5- paragraph 4.1: please consider the use of oxalate-degrading enzymes
6- title of paragraph 4.2: Please call this therapy simply as chaperone therapy because the description is focused on Vitamin B6, that is a pharmacological chaperone , and betaine, that is a chemical chaperone
7- line 423: please cite also clinical cases of defective GO
The English style should be improved.
Comments on the Quality of English LanguageSeveral style and syntax errors are present throughout the manuscript
Author Response
Dear reviewer:
We extend our gratitude for your thorough review and feedback on my manuscript. In accordance with your valuable suggestions and requirements, we have meticulously revised our work. Each modification has been carefully annotated within the answers to indicate its exact location, following a point-by-point response format. Notably, we also engaged an English professor to rigorously edit and refine the language and grammatical structure of the manuscript. We are deeply grateful for your insightful comments, which have significantly contributed to the enhancement of our manuscript. Herein, we address each comment specifically:
- the use of siRNA as new therapeutic option is described at the end of the manuscript although it is cited before in several instances (i.e. lines 215 and 314 where the new drugs are cited without a prior explanation). I would suggest to remove the references and describe the changes in the therapeutic approach in paragraph 4.5.
A:Thanks for your suggestion of the structure of manuscript.
- We relocated the siRNA description from section 4.1, paragraph 1 (original manuscript line 215), to section 4.5, paragraph 2 (revised manuscript lines 456-461), elucidating the impact of such drugs on traditional nocturnal hydration therapy.
- In section 4.3's dialysis segment (original manuscript line 314), RNAi is merely cited in parentheses as an example of therapies reducing oxalate, not as a novel drug reference. Additionally, the primary purpose of the sentence is to emphasize the importance of early dialysis, so we consider it more appropriately placed in the 4.3 section (revised manuscript line 328).
- Furthermore, upon review, RNAi drug therapy is mentioned in section 4.4's surgical intervention, paragraph 2. This section illustrates the possibility that CLKT and SLKT have been gradually replaced by IKT in recent years, and placing it in that section may be more appropriate and facilitate guidance for clinical physicians. Moving this to section 4.5 might imply a lack of innovation in chapter 4.4's transplantation discourse, conflicting with our review's objective. After careful consideration, we decided to leave the treatment with RNAi drugs in chapter 4.4 unchanged for the time being (revised manuscript lines 382-391).
- the format of table 1 should be improved.
A: Thank you for your recommendations regarding Table 1's details. The journal editor altered Table 1's format, deviating from the initially submitted version. We have thus reformatted the table to enhance its readability (revised manuscript line 51).
- line 132: please check the actual number of mutations.
A: Thank you for the detailed suggestions. The exact mutation count was verified through the HGMD website (http://www.hgmd.cf.ac.uk), consistent with existing literature, and the referenced literature has been updated accordingly (revised manuscript lines 128-131) [1,2].
- line 178: I would not consider SLC26A6 mutation as a form of primary hyperoxaluria.
A: Thank you very much for your comments and suggestions to revise the article. Although intestinal wall-associated genetic changes (e.g., SLC26A6) are one of the etiologic factors contributing to hyperoxaluria, they should not be classified as primary hyperoxaluria. We have already made the necessary corrections in the manuscript (revised manuscript lines 168-176).
- paragraph 4.1: please consider the use of oxalate-degrading enzymes.
A: We appreciate the contributions to this manuscript. Indeed, the spectrum of enzyme administration therapies encompasses both enzyme replacement therapy (ERT) and enzyme substitution therapy (EST), the latter being delineated in Section 4.6.2. Whereas the development of such drugs in EST has been aimed at patients with oxalosis and secondary hyperoxaluria, it has been added to the section 4.1 given its suggestive role in the treatment of PH (revised manuscript lines 250-267). In fact, EST has rarely been used in clinical trials for PH patients, with published information indicating less-than-ideal outcomes or undisclosed results post-completion.
- title of paragraph 4.2: Please call this therapy simply as chaperone therapy because the description is focused on Vitamin B6, that is a pharmacological chaperone , and betaine, that is a chemical chaperone.
A: We express our gratitude for the revisions applied to our article. The heading of Section 4.2 has been updated to "Chaperone Therapy" to enhance accuracy (revised manuscript line 287).
- line 423: please cite also clinical cases of defective GO.
A: We sincerely appreciate your meticulous review and enhancement recommendations for the article. Clinical case reports on GO defects are exceptionally scarce, and we have added representative cases accordingly (revised manuscript lines 433-436) [3].
References
- Dindo, M.; Mandrile, G.; Conter, C.; Montone, R.; Giachino, D.; Pelle, A.; Costantini, C.; Cellini, B. The ILE56 Mutation on Different Genetic Backgrounds of Alanine:Glyoxylate Aminotransferase: Clinical Features and Biochemical Characterization. Mol. Genet. Metab. 2020, 131, 171–180, doi:10.1016/j.ymgme.2020.07.012.
- Wu, J.; Song, J.; He, Y.; Zhong, C.; Yang, Q.; Li, Q.; Wang, M. rolithiasis 2023, 51, 123, doi:10.1007/s00240-023-01494-8.
- Frishberg, Y.; Zeharia, A.; Lyakhovetsky, R.; Bargal, R.; Belostotsky, R. Mutations in HAO1 Encoding Glycolate Oxidase Cause Isolated Glycolic Aciduria. J Med Genet 2014, 51, 526–529, doi:10.1136/jmedgenet-2014-102529.

Reviewer 2 Report
Comments and Suggestions for Authors
Review of the manuscript biomolecules-2949750
Navigating the Evolving Landscape of Primary Hyperoxaluria: Traditional Management Defied by the Rise of Novel Molecular Drugs
By Yueqi Huang et al.
The evaluated paper is a review article. After a short Introduction containing the classification of primary hyperoxaluria, the Authors characterize the biochemical basis of the described disorder and present a diagnostic and therapeutic protocol illustrating the recommended treatment for hyperoxaluria. In the further part of the review, the Authors describe the currently used pharmacological treatment, briefly characterize the role of dialysis in oxalate clearance and surgical options for removing oxalate kidney stones. The final part contains a description of newly developed RNAi therapies, as well as the possibilities currently in preclinical stages, based on viral vectors (adenovirus or lentivirus) or non-viral delivery systems (such 529 as lipofectamine and cell-penetrating peptides) to deliver clustered regularly interspaced 530 palindromic repeats (CRISPR)/Cas9. Moreover, an enzyme replacement therapy (ERT) is also described.
The article has a coherent and well-thought-out plan, and the text is illustrated with two figures.
I think that before the final publication, the authors should consider making possible corrections:
1. In subchapter 3 (Diagnosis and Genetic Characterization), it is worth mentioning genetic counseling and prenatal/preimplantation tests for pregnant women at risk of developing hyperoxaluria and among families with this disorder
2. Figure 2 - due to the fact that the treatment modalities are essentially similar in both diagnostic paths, can they be common, unified, with an indication of any differences?
3. In the subsection 4.1. it is worth describing additionally/more broadly the therapy with alkaline citrates, orthophosphates and magnesium (oxalate crystallization inhibitors)
4. The fragment regarding the content of oxalates in the diet in various foods (lines 227-233) is worth providing a table giving the detailed content of oxalates in foods very rich in these compounds
5. In the fragment regarding treatment, in subsection 4.1. - when discussing the role of bacteria, please refer to Enterococcus faecalis - this bacterium also has enzymes that degrade oxalic acid - is it also used or tested for use in disturbed oxalate metabolism?
6. It is also worth adding to the text a section devoted to inhibitors of hepatic oxalic acid synthesis
Author Response
Dear reviewer:
We extend our gratitude for your thorough review and feedback on my manuscript. In accordance with your valuable suggestions and requirements, we have meticulously revised our work. Each modification has been carefully annotated within the answers to indicate its exact location, following a point-by-point response format. Notably, we also engaged an English professor to rigorously edit and refine the language and grammatical structure of the manuscript. We are deeply grateful for your insightful comments, which have significantly contributed to the enhancement of our manuscript. Herein, we address each comment specifically:
- In subchapter 3 (Diagnosis and Genetic Characterization), it is worth mentioning genetic counseling and prenatal/preimplantation tests for pregnant women at risk of developing hyperoxaluria and among families with this disorder.
A: We are grateful for your thorough review of our manuscript. The significance of preconception counseling and prenatal/preimplantation testing in high-risk couples with primary hyperoxaluria is highlighted in the concluding paragraph of Section 3, supported by references to pertinent literature (revised manuscript lines 187-190) [1,2].
- Figure 2 - due to the fact that the treatment modalities are essentially similar in both diagnostic paths, can they be common, unified, with an indication of any differences?
A: We appreciate your expert recommendations. Initially, we attempted to simplify the imagery as suggested; however, combining two distinct treatments in a single box with annotations obscured their differences, complicating comprehension, particularly for non-specialists. Consequently, we retained the latest version of Figure 2, prioritizing the depiction's clarity from diagnosis to treatment (revised manuscript lines 123-127). Despite the similarities in treatment approaches for hyperoxaluria and hyperoxalemia, contrasting the two boxes enhances their differentiation, offering clinicians and researchers a more intuitive understanding.
- In the subsection 4.1. it is worth describing additionally/more broadly the therapy with alkaline citrates, orthophosphates and magnesium (oxalate crystallization inhibitors).
A: We highly value your recommendations for enhancing the article's content and express our profound gratitude for your thoughtful review. In subsection 4.1, we have comprehensively discussed both treatment modalities, namely alkaline citrate and orthophosphate (revised manuscript lines 212-222). And an additional introduction on magnesium has been added (revised manuscript lines 269-276).
- The fragment regarding the content of oxalates in the diet in various foods (lines 227-233) is worth providing a table giving the detailed content of oxalates in foods very rich in these compounds.
A: We are deeply grateful for your insights regarding our manuscript. First, the expert consensus indicates that a low-oxalate diet is not essential for patients with primary hyperoxaluria, suggesting that detailing the oxalate content of foods in a separate table may detract from the article's core focus [3]. Second, variations in the oxalate content of foods, attributed to analytical methods and biological differences (e.g., cultivar, harvest time, and growing conditions), lead to substantial discrepancies without consensus [4]. Moreover, significant variability in oxalate absorption rates across populations for different foods suggests that avoiding foods with high or very high oxalate content is a more straightforward approach than detailed meal assessment and documenting daily intake in milligrams [5]. Considering these factors collectively, we feel that the addition of this table is not appropriate in this review, and we hope you will understand.
- In the fragment regarding treatment, in subsection 4.1. - when discussing the role of bacteria, please refer to Enterococcus faecalis - this bacterium also has enzymes that degrade oxalic acid - is it also used or tested for use in disturbed oxalate metabolism?
A: We are grateful for your thoughtful reminder and inquiry. Enterococcus faecalis, isolated from human feces, has demonstrated oxalate degradation capacity in vitro, but this capacity needs to be maintained under poor nutritional conditions and repeated passaging cultures [6]. The potential for clinical application of this bacterium as a probiotic is highly constrained, compounded by the absence of further research linking Enterococcus faecalis to hyperoxaluria [7,8]. After thorough consideration, we have not included Enterococcus faecalis in the section discussing the role of bacteria. Your understanding of this decision is highly appreciated, and we thank you once more for your insightful feedback on our manuscript.
- It is also worth adding to the text a section devoted to inhibitors of hepatic oxalic acid synthesis.
A: Currently, inhibition of hepatic oxalic acid synthesis primarily relies on RNA interference techniques targeting key enzymes in the oxalate synthesis pathway, as extensively discussed in Section 4.5 of the article (revised manuscript lines 419-537). This includes drugs such as Lumasiran (Oxlumo™), Nedosiran, and Stiripentol. Furthermore, the exploration of CRISPR/Cas9 technology in preclinical therapeutic applications has been addressed in Section 4.6.1 of the manuscript (revised manuscript lines 539-567).
References
- Méaux, M.-N.; Sellier-Leclerc, A.-L.; Acquaviva-Bourdain, C.; Harambat, J.; Allard, L.; Bacchetta, J. The Effect of Lumasiran Therapy for Primary Hyperoxaluria Type 1 in Small Infants. Pediatr Nephrol 2022, 37, 907–911, doi:10.1007/s00467-021-05393-1.
- Marchetti, F.; Corsello, G. Genetics and"democracy". Ital J Pediatr 2022, 48, 202, doi:10.1186/s13052-022-01391-7.
- Groothoff, J.W.; Metry, E.; Deesker, L.; Garrelfs, S.; Acquaviva, C.; Almardini, R.; Beck, B.B.; Boyer, O.; Cerkauskiene, R.; Ferraro, P.M.; et al. Clinical Practice Recommendations for Primary Hyperoxaluria: An Expert Consensus Statement from ERKNet and OxalEurope. Nat Rev Nephrol 2023, 19, 194–211, doi:10.1038/s41581-022-00661-1.
- Massey, L.K. Food Oxalate: Factors Affecting Measurement, Biological Variation, and Bioavailability. J Am Diet Assoc 2007, 107, 1191–1194; quiz 1195–1196, doi:10.1016/j.jada.2007.04.007.
- Attalla, K.; De, S.; Monga, M. Oxalate Content of Food: A Tangled Web. Urology 2014, 84, 555–560, doi:10.1016/j.urology.2014.03.053.
- Hokama, S.; Honma, Y.; Toma, C.; Ogawa, Y. Oxalate-Degrading Enterococcus Faecalis. Microbiol Immunol 2000, 44, 235–240, doi:10.1111/j.1348-0421.2000.tb02489.x.
- Wigner, P.; Bijak, M.; Saluk-Bijak, J. Probiotics in the Prevention of the Calcium Oxalate Urolithiasis. Cells 2022, 11, 284, doi:10.3390/cells11020284.
- Murru, N.; Blaiotta, G.; Peruzy, M.F.; Santonicola, S.; Mercogliano, R.; Aponte, M. Screening of Oxalate Degrading Lactic Acid Bacteria of Food Origin. Ital J Food Saf 2017, 6, 6345, doi:10.4081/ijfs.2017.6345.
Please also see the attachment.

Round 2
Reviewer 1 Report
Comments and Suggestions for Authors
The authors addressed all the raised concerns. The manuscript is now suitable for publication.